# Mediating and Moderating Effects of Family Cohesion between Positive Psychological Capital and Health Behavior among Early Childhood Parents in Dual Working Families: A Focus on the COVID-19 Pandemic

**DOI:** 10.3390/ijerph18094781

**Published:** 2021-04-29

**Authors:** In Young Cho, Sun-Hee Moon, Ji Yeong Yun

**Affiliations:** 1College of Nursing, Chonnam National University, 160 Baekseo-ro, Dong-gu, Gwangju 61469, Korea; kikiin1024@jnu.ac.kr; 2Department of Nursing, Jesus University, 383 Seowon-ro, Wansangu, Jeonju-si 54989, Jeollabukdo, Korea; yunjiyeong0221@gmail.com

**Keywords:** early childhood, health promotion behavior, dual working parents, COVID-19, family cohesion

## Abstract

This study examined the mediating and moderating effects of family cohesion on the relationship between positive psychological capital and health promotion behaviors among dual working parents of young children during the COVID-19 pandemic. We collected data from 214 dual working parents and analyzed the results using the IMB SPSS version 26.0 software. We found that positive psychological capital had significant effects on both family cohesion (β = 0.19, *p* < 0.001) and health promotion behaviors (β = 0.26, *p* < 0.001), while family cohesion showed a significant mediating effect on health promotion behaviors (β = 0.34, *p* = 0.002). Positive psychological capital (independent variable) also had significant effects on health promotion behaviors (β = 0.19, *p* < 0.001). Finally, a Sobel test showed that family cohesion had a significant partial mediating effect on the relationship between positive psychological capital and health promotion behaviors (Z = 2.70, *p* = 0.005) but no moderating effect. Thus, it is necessary to enhance health promotion behaviors through programs focused on both family cohesion and positive psychological capital. However, the fact that participants in this study were only from South Korea highlights the requirement for future research that includes participants from different cultural contexts and social systems.

## 1. Introduction

The coronavirus disease 2019 (COVID-19) outbreak, which was first identified in Wuhan, China at the end of 2019, has spread around the world and has had a tremendous impact on health and quality of life of all populations, including children and their parents [1,2,3,4].

Worldwide, the COVID-19 infection rate among children has been constantly increasing, accounting for 11.0% of all infections as of April 2021 and as this group is largely asymptomatic and displays variable symptoms (i.e., asymptomatic to severe pneumonia), they are likely to remain undiagnosed and spread it to others [4,5,6,7,8]. In particular, young children aged 1–6 have relatively weak immunity and are less capable of coping with external infectious agents than adults. Moreover, they are especially prone to breathing difficulties and have a very high risk of respiratory disease [9,10]. Based on the above fact, young children are susceptible to both infection and reinfection with COVID-19. In addition, after infection with COVID-19, risk of complications such as multisystem inflammatory syndrome in children can also increase [11].

Although several vaccines for COVID-19 have been approved recently, and some countries have begun vaccinating health workers as well as high-risk groups, these vaccines have only been approved for use in adults, and vaccinations for children and adolescents are still in development [12]. In this crisis, children and their families are experiencing unprecedented challenges and various difficulties [13]. Any potential sequelae could ultimately affect their health, growth, and development; thus, it is important to prioritize this population.

However, worldwide, there has been limited research on pediatric COVID-19. The focus has been mainly on adults and the elderly, since the onset of the COVID-19 epidemic; children have been less affected than adults in terms of severity and frequency of the disease [14]. Nevertheless, it is still important for children to continue taking precautions, including wearing masks, regular handwashing, and physical distancing, to control the spread of the virus. Furthermore, a long-term impact of the pandemic on promoting children’s health or health behavior will be inevitable [3].

As COVID-19 can greatly impact overall health due to its persistent nature, early health promotion behavior can greatly improve long lasting effects on overall health and growth across all areas of development [15,16]. Health promotion behavior is a motivated action to realize health potential in order to reach an optimal state of health. As such, in COVID-19 situations, this includes not only direct actions to prevent infection but also a set of complex behaviors aimed at promoting long-term growth and development (e.g., safety, disease prevention, emotional support, activity and rest, hygiene, and physical activity) in a wide range of people [3,17]. Since anyone can be infected with COVID-19, early health promotion behavior is a very important task not only for children suffering from diseases but for all children and parents.

Worldwide, parents serve as primary care providers and exert the greatest influence on the health promotion behavior of their children. Indeed, parental influence works as a foundation for healthy child growth and development [17,18].

Dual Working Parents and COVID-19

Many working parents (who engage in paid work while also caring for their children at home) have been experiencing a variety of difficulties and stress resulting from COVID-19 [18,19]. There have been two waves of COVID-19 in South Korea, and the government adopted an emergency response starting on 23 February 2020 in order to prevent the spread of the infection [4]. All schools, both at national and local levels, were closed; social distancing became an important intervention, and the movement of people was reduced drastically [20,21]. Under these conditions, parents are increasingly playing pivotal roles at the forefront of protecting children from this infection and safeguarding their health and well-being [13]. However, previous research has shown that dual working parents exhibit lower health behaviors than single working parents [22]. Furthermore, as both parents must work in this arrangement, children must be cared for elsewhere (i.e., emergency child care institution), which may expose both parents and children to frequent contact with other people, thereby increasing their risk of exposure to the virus. Parents feel uncertainty regarding when and how they might be infected and anxiety due to the delay in vaccine development, which further aggravates these feelings [23].

Based on the above, we assume that, particularly during the COVID-19 pandemic, more focus should be placed on dual working parents for the prevention of infection and health promotion of their children. This highlights the necessity for programs aimed at increasing health promotion behaviors among working parents of young children [9,24].

Previous studies have emphasized the necessity to examine various environmental factors (e.g., family-related and individual factors) based on an integrated understanding of their health promotion behaviors [15,17]. Individual factors include positive psychological capital, which works as an internal force that can help overcome problems while increasing the ability to adapt to family stress; this includes self-efficacy, hope, resilience, and optimism. Numerous research has found it to be related to health promotion behavior of individuals [25,26,27]. As such, a series of studies on variables related to positive psychological capital and health promotion behavior have been conducted [27,28]. However, Koh and Park (2019) [29] argued that positive psychological capital can affect psychological well-being and health through the influence of social environmental factors, rather than having a direct influence. Therefore, it is necessary to investigate which factors could be used to increase health behavior.

An overview of related studies led us to assume the family-related cohesion factor as a third variable that mediates the relationship between positive psychological capital and health promotion behavior. Family cohesion is an environmental variable that refers to the emotional bonds shared by family members [30]. Cho [31] and Kim and Kwon [32] suggested that by receiving sufficient affection and support from family, parents can place more interest and have a positive effect on their children’s health promotion activities; this is all the more important during the COVID-19 pandemic [3]. However, there is a limited number of studies of how family cohesion (which is an important concept in overcoming the COVID-19 pandemic) plays a role in the relationship between positive psychological capital and health promotion behavior. In addition, previous research on the mediating and moderating effects of family cohesion has focused mainly on negative aspects, such as family burden, social isolation, depression, and stress [33,34,35]. Therefore, further research is required to promote the health promotion activities of dual working parents, focusing on the positive health behaviors of children and the variables that affect their development. Considering that parents and family-related factors are key factors in health promotion, it is necessary to identify how family cohesion influences parents’ health promotion to cope with long-lasting COVID-19.

Another important reason to examine the family cohesion variable arises from the fact that positive psychology capital does not necessarily lead to health-related behavior [36]. Yet, to the best of our knowledge, the above conflicting results on the role of mediating or moderating family cohesion support scholars’ argument that the effects of positive psychological capital on health promotion behaviors are not homogenous.

Therefore, to more accurately identify the relationship between the predictor variable and dependent variable, it is first necessary to clarify how the positive psychological capital of dual working parents affects health promotion behavior, and under what conditions the health promotion behavior can be changed through the family cohesion factor. Moreover, considering differences in health among dual working parents who must care for children both at work and at home during COVID-19, related research is both limited and outdated, with most focusing on mental health status and stress rather than overall health promotion behaviors [37,38].

Based on this theoretical background, the present study examines the mediating effect and also the moderating effect, which suggests that health promotion behavior might vary according to the level of family cohesion among dual working parents with young children during the COVID-19 pandemic. Moreover, to date, most studies on the health promotion behaviors of parents have failed to comprehensively explain the differences between fathers and mothers and have primarily targeted mothers due to their traditionally important roles in childcare, an idea that stems from Confucian ideologies in many Eastern cultures [19]. However, fathers are now playing more integral roles in this regard, particularly as a result of modern demands, thus highlighting the requirement to further examine their participation in parenting [39].

Our results constitute basic data for use in the development of a very timely and effective health promotion program aimed at children and their families. The hypotheses of the study are as follows (Figure 1).

**Hypotheses** **H1.**
*Family cohesion mediates the relationship between positive psychological capital and health promotion behavior.*


**Hypotheses** **H2.**
*Family cohesion has a moderating effect on the relationship between positive psychological capital and health promotion behavior.*


## 2. Materials and Methods

### 2.1. Participants

Participants included the parents of young children (1–6 years of age: 12–72 months) in South Korea. The specific selection criteria were as follows: (1) parents who understand the contents of the study and can complete the questionnaire; (2) parents of children who do not have congenital disorders, diseases with reduced immune function, or chronic diseases. All participants confirmed their understanding of the study purpose and agreed to participate on a voluntary basis through online written consent. For this study, we determined that the minimum sample size was 194 based on our inclusion of 14 predictor variables (effect size of 0.15, alpha of 0.05, and power of 0.95; regression analysis using G Power 3.1.9.7). This study is a secondary analysis study of [IRB No. 2-7008161-A-N-01]. Of the 432 participants recruited in the original survey [IRB No. 2-7008161-A-N-01], parents who answered they were dual working were included. After excluding respondents who either did not complete their consent forms or submitted improper responses, we collected 214 valid responses, which was more than adequate. However, this study has the limitation of including participants only from South Korea and not from other different cultural contexts, social systems, and perceptions about childcare, during the COVID-19 pandemic, thereby suggesting further scope for comprehensive research.

### 2.2. Procedures

This study was approved by Chonnam National University Institutional Review Board (1040198-210127-HR-013-01). All procedures performed in the studies were in accordance with the ethical standards of the institutional and/or national research committee.

The data collection was conducted by online-based survey (which was safer than engaging in face-to-face surveys) between 10 November and 5 December 2020 in South Korea. This coincided with the second wave of COVID-19 in the country, which infected an average of more than 500 people per day after the first wave of the pandemic in March 2020. In this period, the number of infected people per day showed a rapid increase, and consequently, the operation of public facilities was stopped, and the government upgraded social distancing measures.

The research announcement was promoted in three social media platforms (parenting-related communities), where around 50,000 out of 50 million parents in South Korea, from all regional backgrounds, share information and opinions about their children’s health and care. Participants originated from seven geographical regions, which is enough to consider it representative of the parents in South Korea. We provided respondents with sufficient information about the study purpose and all related procedures. Furthermore, we also asked them in advance to express their state of mind and behavior in the current COVID-19 situation. For participants with two or more 1–6-year-old children, we asked them to respond based on the first child. We also assured participants regarding the anonymity and confidentiality of their responses and asserted that any collected data would only be used for research purposes. Participants were told they could withdraw from the study at any time if desired. Voluntary parents who gave approval to participate answered the questions. All participants were given small gifts as expressions of gratitude.

### 2.3. Measures

#### 2.3.1. Positive Psychological Capital

We used the Positive Psychological Capital Scale developed by Luthans, Youssef, and Anolio [27] and modified by Lim [40]. All items were answered on a 5-point Likert scale ranging from 1 (not at all) to 5 (very much), with possible scores ranging from 18 to 90 points (average rating 1 to 5 points). Here, higher scores indicate higher levels of positive psychological capital. Lim’s [40] study showed a Cronbach’s α value of 0.93, while this study showed a value of 0.95.

#### 2.3.2. Family Cohesion

We measured family cohesion using 10 relevant items from the Family Adaptability and Cohesion Evaluation Scale (FACES III), which was developed by Olsen, Porter, and Lavee [41]) and translated by Ahn [42]. All items were rated on a 5-point Likert scale ranging from 1 (almost never) to 5 (always), with possible scores ranging from 10 to 50 (average score of 1 to 5 points). Here, higher scores indicate stronger family cohesion. Olson, Porter, and Lavee (1985) showed a Cronbach’s α value of 0.77, while this study showed a value of 0.84 [41].

#### 2.3.3. Health Promotion Behaviors

We modified the measuring tool for health promotion behaviors previously developed by Cho [31] and revised by Ki [43] for the current situation of COVID-19, according to the purpose of our research, to understand the health promotion behavior of parents during the COVID-19 pandemic. This is a tool that measures the prevention of infection and the promotion of long-term health care in the COVID-19 pandemic. It was revised based on the guidelines of Choi (2015), CDSCH (2020), and WHO (2020) [44,45,46].

The revised questions were verified for content validity by two professors of pediatric nursing, two professors of nursing, and one pediatrics specialist, which was S-CVI/Ave = 0.96. In addition, a preliminary survey was conducted among 10 parents of young children in order to assess the readability and appropriateness of the questions. Finally, the tool consists of 29 items, each rated on a 4-point Likert scale ranging from 1 (not at all) to 4 (always). Here, higher scores indicate better health promotion behaviors (i.e., a higher degree of health promotion). Cho (2004) [31] showed a Cronbach’s ⍺ value of 0.86, while this study showed a value of 0.85.

### 2.4. Data Analysis

The collected data were analyzed using IBM SPSS Statistics 26.0 (IBM Corp., Armonk, NY, USA). First, we examined the general participant characteristics, positive psychological capital, family cohesion, and health promotion behaviors based on descriptive statistics (percentages, means, and standard deviations). Next, we analyzed the differences between positive psychological capital, family cohesion, and health promotion behaviors based on the general characteristics; this was performed using the independent t-test and one-way ANOVA, while a post hoc analysis was performed using the Scheffé test. Further, we assessed the correlations between variables via Pearson’s correlation coefficients. We used hierarchical regression analysis to test the mediating effects of family cohesion on the relationship between positive psychological capital and health promotion behaviors suggested by Baron and Kenny (1986) [47], followed by an additional Sobel test. Finally, we evaluated the moderating effect via hierarchical regression analysis suggested by Cohen, West, and Aiken (2013) [48], who argued that the utilization of regression through interaction effects can also be used to verify moderating effects, if the predictor is a numerical variable [49].

## 3. Results

### 3.1. Health Promotion Behaviors Based on General Characteristics

Table 1 shows general participant characteristics and health promotion behaviors. The average participant age was 36.49 ± 3.46 years (range of 27–45 years); there were 165 females (77.1%).

As for family structure, the vast majority of participants lived in nuclear arrangements (199; 92.99%), followed by large families (13; 6.07%); the highest proportion of participants had one child (112; 52.34%), followed by those with two (95; 44.39%). For the children, the average age was 44.14 ± 19.37 months (range of 12–72 months). Further, most parents had not experienced self-isolation or acquaintance COVID-19 infection experience. We found significant differences in health promotion behaviors based on the participant characteristics, including age (F = 76.55, *p* < 0.001) and gender (t = −3.22, *p* = 0.002). Regarding gender, females (M = 103.60, SD = 7.81) showed higher health promotion behaviors than males (M = 98.69, SD = 9.88) (Table 1).

### 3.2. Positive Psychological Capital, Family Cohesion, and Health Promotion Behaviors

Table 2 provides information on participants’ health promotion behaviors, positive psychological capital, and family cohesion among dual working parents. The average positive psychological capital was 66.74 ± 10.78 of 90 possible points, while family cohesion was 40.6 ± 5.55 of 50 possible points, and health promotion behavior was 102.5 ± 8.56 of 116 possible points (Table 2).

### 3.3. Relationships between Health Promotion Behaviors and Positive Psychological Capital/Family Cohesion

Table 3 shows the relationships between health promotion behaviors and related factors among participants. As shown, health promotion behaviors were positively correlated with positive psychological capital (r = 0.32, *p* < 0.0001) and family cohesion (r = 0.31, *p* < 0.0001). There was also a positive correlation between positive psychological capital and family cohesion (r = 0.37, *p* < 0.0001) (Table 3).

### 3.4. The Mediating Effect of Family Cohesion on the Relationship between Positive Psychological Capital and Health Promotion Behaviors

Prior to verifying the mediating effect, we tested for multicollinearity and variable independence, thus revealing a Durbin–Watson index of 1.83 (showing independence); the tolerance limit for the variable was 0.87 (i.e., more than 0.1). In addition, the variance inflation factor index was 1.15, which did not exceed 10, thus indicating no problems with multicollinearity between the independent variables. Ultimately, the regression model was suitable due to satisfactory results for both the residual normality distribution and equal variance.

As mentioned earlier, we implemented the method proposed by Baron and Kenny (1986) to test the mediating effect of family cohesion. In step 1 of the regression analysis, positive psychological capital (independent variable) had a significant effect on health promotion behaviors (β = 0.26, *p* < 0.001) (explanatory power of 10%). In step 2, our analysis of the effect of the independent variable on the parameter showed that positive psychological capital had a significant effect on family cohesion (β = 0.19, *p* < 0.001) (explanatory power of 13%). In step 3, health promotion behaviors (dependent variable) and both positive psychological capital and family cohesion were simultaneously applied to the model to control for mutual influences. Results showed that family cohesion (independent variable) had a significant mediating effect on health promotion behaviors (β = 0.34, *p* = 0.002). Positive psychological capital (independent variable) also had a significant effect on health promotion behaviors (β = 0.19, *p* < 0.001).

Generally, a mediating variable is considered to have partial mediating power if the relationship between the independent and dependent variable is statistically significant, and the degree of influence of the independent variable on the dependent variable is reduced. In sum, our analysis showed that positive psychological capital had a statistically significant effect on health promotion behaviors, with the effect decreasing to 0.19 compared to the β value of 0.26. As such, family cohesion appeared to have a partial mediating effect. The estimate of the indirect effect was 0.19 × 0.34 = 0.06. We also conducted a Sobel test to verify the significance of the partial mediating effect, thus revealing a significant indirect effect (Z = 2.70, *p* = 0.005) (Table 4) (Figure 2).

### 3.5. The Moderating Effect of Family Cohesion on the Relationship between Positive Psychological Capital and Health Promotion Behaviors

We conducted a regression analysis to test for moderating effects among variables. First, we found that positive psychological capital (independent variable) had a significant effect (β = 0.43, *p* = 0.002) on health promotion behaviors (dependent variable). Second, we found that family cohesion (moderating variable) had a significant effect (β = 19.29, *p* = 0.039) on health promotion behaviors (dependent variable). However, the moderating effect of family cohesion was not statistically significant when the term of interaction between positive psychological capital and family cohesion was added. As such, family cohesion had no moderating effect on the relationship between positive psychological capital and health promotion behaviors (Table 5) (Figure 2).

## 4. Discussion

It is critically important to ensure that parents and their children maintain positive health promotion behaviors, particularly due to the continuous spread of COVID-19. Thus, it is urgent to establish a health promotion plan to help dual working parents. For this purpose, we attempted to identify the mediating and moderating effects of family cohesion on the relationship between positive psychological capital and health promotion behaviors among dual working parents during the COVID-19 pandemic.

In general, we found that mothers tend to have better health promotion behaviors than fathers. In other words, although more women have jobs now than in the past, a long-standing traditional family arrangement still persists, in which women are seen as more responsible for childcare in South Korea [38,50]. However, if most children’s health care responsibility is placed on the mother, in the situation that children are unable to attend normal school and have to stay at home during the COVID-19 pandemic, this can also be a burden for working women if it lasts for a long time. This indicates that fathers’ participation in caring and health behavior should be given more attention and encouraged among dual working populations. As such, Kim [51] found that higher levels of spousal support were associated with better child health. As recent COVID-19 trends are encouraging fathers to increase their participation in childcare, coparenting is also a significant factor [52,53]. In addition, institutional support in the form of family friendly systems that allow both mother and father to take parental leave (taking time off from work to help with childcare), and more flexible working conditions (i.e., telecommuting working) are required. In such an arrangement, parents can take more time to care for and educate their children about health behaviors, including hygiene management, eating habits, and indoor physical activity, especially, as children are experiencing difficulties both with risk of COVID-19 infection and lack of outdoor activity during the pandemic.

We verified the mediating effect of family cohesion on the relationship between positive psychological capital and health promotion behaviors; the addition of family cohesion during the third stage of verification reduced the influence of positive psychological capital on health promotion behaviors, thus supporting the partial mediating effect of family cohesion. This is partially similar to the results of a study by Bae et al. [54], in which family functions, such as family cooperation, affection, and intimacy, were shown as partial mediating variables in the relationship between depression and health promotion behaviors.

During the COVID-19 pandemic, dual working parents are experiencing higher stress and anxiety related to the notion that they should pay more attention to their children’s care and health (infection prevention behavior, eating habits, physical activity, etc.), since their children must be left to emergency care institutions due to the closures of daycare centers and kindergartens. However, individuals with higher positive psychological capital can effectively overcome stress and negative emotions caused by COVID-19, thus demonstrating their coping skills via positive psychological assets [55,56]. As positive psychological capital is a concept addressing human strengths and positive aspects [56,57], we can judge that the positive perceptions of parents themselves will influence cohesion when interacting with their own family members. In turn, this strengthens the family bond, as the family becomes united in their goal to cope with the confusing COVID-19 situation.

Luthans [57] argued that positive psychological capital can be developed through learning or training and can be changed based on individual development efforts. This suggests the necessity for nursing interventions that enhance positive psychological capital, which gives strength to overcome COVID-19 situations; for example, implementing counseling/education programs and institutional support for parents by establishing a synergy between self-efficacy, hope, optimism, and resilience. Here, it is mostly necessary to instill positive expectations and beliefs among parents by, for instance, assuring them that the future may be better than the present pandemic.

Based on the above results, we can also determine that family cohesion will ultimately have a positive mediating effect on health promotion behaviors for both parents and children. It can bring all family members together and help them to more actively engage in infection prevention while developing better beliefs and attitudes toward health promotion behaviors [58]. This means that family factors, such as family cohesion, which was a major variable in this study, must be included both directly and indirectly in the context of family-centered health promotion strategies in the COVID-19 pandemic.

In some respects, we can look forward to the positive aspect that South Korean families, who do not normally engage in extended family sharing time as result of long working/school hours, will spend more time with their families due to the government’s social distancing measures in response to the COVID-19 pandemic.

In family environments, especially where family members find it difficult to allocate time to each other (such as dual working family), local and national efforts are required to encourage families to participate in non-face-to-face online programs, such as physical activities, nutrition, and providing health information for all members of the family. In addition, as many studies have reported the risk of obesity in children due to outside and physical activity constraints in the COVID-19 epidemic, it is necessary to make family time to participate in exercises, such as climbing stairs, jumping rope, indoor stretching, and yoga for all family members, thereby promoting communication and interaction between parents and children. In addition, hands-on activities, and family joint games to create a healthy family culture, can also be helpful [59].

In order to limit the spread of COVID-19, the whole family should be united, and parents should be great role models for children. For example, if parents wash their hands frequently, keep at least six feet away from others, and protect themselves and others by wearing masks in public, children are likely to behave in the same way. If parents show healthy eating habits, obtain enough sleep, participate in exercise or relaxation training, and let children see and follow them, children are likely to want to follow these behaviors. Furthermore, creating and sharing responsibilities with other trusted guardians (e.g., childcare caregivers) who practice social distancing and hygiene measures can provide parents time to deal with their stress and take health measures.

In addition, at the national level, in order to actively cope with the environment changed by COVID-19, it is also necessary to develop online programs and smartphone applications that parents and their children can use regardless of time and space constraints, including monitoring daily health activities, while maintaining family bonds and child–parents attachments.

However, this study did not find a moderating effect of family cohesion between positive psychological capital and health promotion behaviors. In other words, family cohesion had the most powerful effect on health promotion behaviors, but no buffering effect was found, suggesting that extremely high and low family cohesion does not change the effect of positive psychological capital on the health behavior. In particular, this study result cannot rule out the possibility that data collection was made at the time of the pandemic. In addition, we can assume that the factors affecting the health promotion behaviors were out of control, as various factors can affect health promotion behavior, such as self-efficacy, stress, parental attachment, and epidemic environmental factors [9,27]. Thus, future studies are required that consider the various variables affecting the progress of health behavior in COVID-19 situations.

Despite the results, this study is meaningful to understand the importance of ensuring and maintaining an adequate family cohesion. Therefore, it is necessary to study the results repeatedly during the rapidly changing COVID-19 period and compare and analyze those results to comprehensively understand the role of family cohesion in the COVID-19 epidemic. It is also necessary to clearly identify the influence of family cohesion by conducting various investigations related to the influence of health promotion behaviors based on the level of family cohesion.

The findings from our study might have theoretical implications for developing potentially effective family-centered interventions on the promotion of a wide range of health behaviors for dual working parents in the context of COVID-19, which has substantially impacted society as a whole.

## 5. Conclusions

This study was conducted to help improve health promotion behaviors among dual working parents by verifying the mediating and moderating effects of family cohesion on the relationship between positive psychological capital and health promotion behaviors. We found that family cohesion had a partial mediating effect in this regard; however, its moderating effect was not supported. Our results are meaningful in that they not only identified the relationship between positive psychological capital and health promotion activities but also specifically found that family cohesion should be implemented in strategies designed to increase health promotion behaviors for parents. To enhance health promotion behaviors during the stressful events of COVID-19, programs should therefore be developed in consideration of both positive psychological capital and family cohesion.

For future research, we suggest the following. First, this study was limited to participants in South Korea, which means it is necessary to conduct research in different cultural contexts with different social systems and perceptions about childcare, particularly among dual working parents during the COVID-19 pandemic. This will provide a basis for comparison. Second, this study did not target both parents in each family, which makes it necessary to identify differences between mothers and fathers in regard to health promotion behaviors, even in the same family context. It is also necessary to compare health behaviors between dual working parents and those with other arrangements. Finally, it is necessary to conduct a longitudinal study by developing programs that can increase family cohesion and health promotion behaviors among dual working parents, thus providing a way to determine whether these factors affect the rate of infection stemming from COVID-19.

## Figures and Tables

**Figure 1 ijerph-18-04781-f001:**
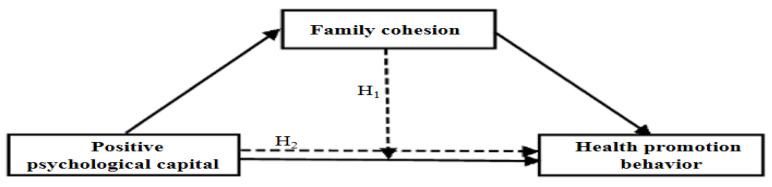
Conceptual framework.

**Figure 2 ijerph-18-04781-f002:**
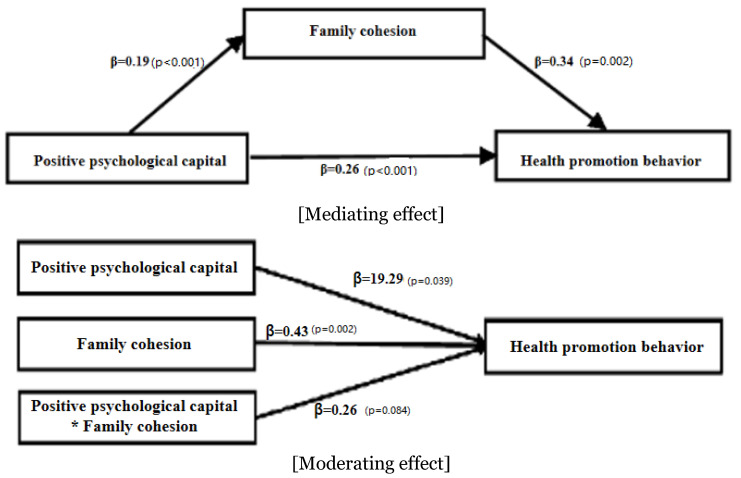
Mediating and moderating effects of family cohesion on the relationship between positive psychological capital and health promotion behavior.

**Table 1 ijerph-18-04781-t001:** General participant characteristics and health promotion behavior (*n* = 214).

Characteristics	Categories	Dual Working Parents Health Promotion Behaviors		
		n (%)	M ± SD	t or F (*p*)
Age (years)	20–29 ^a^	3 (1.40)	112.33 ± 0.58	76.55 (<0.001 **) a > c
	30–35 ^b^	73 (34.11)	103.92 ± 9.19	
	36–40 ^c^	112 (52.34)	101.52 ± 8.44	
	>40 ^d^	26 (12.15)	101.69 ± 6.60	
Gender	Male	49 (22.90)	98.69 ± 9.88	−3.22 (0.002 **)
	Female	165 (77.10)	103.60 ± 7.81	
Family structure	Nuclear family	199 (92.99)	102.70 ± 8.36	0.55 (0.648)
	Large family	13 (6.07)	99.85 ± 11.72	
	Single-parent family	1 (0.47)	104.00 ± 0.00	
	Etc.	1 (0.47)	98.00 ± 0.00	
Child age (month)	12–36	86 (40.19)	102.94 ± 8.18	1.62 (0.201)
	37–59	61 (28.50)	103.56 ± 8.25	
	60–72	67 (31.31)	101.00 ± 9.22	
COVID-19 infection experience	Yes	8 (3.74)	99.50 ± 6.90	−1.25 (0.211)
	No	206 (96.26)	102.70 ± 8.63	
Self-isolation experience	Yes	38 (17.8)	101.70 ± 9.48	−1.21 (0.227)
	No	176 (82.2)	103.16 ± 8.21	
Health status	Healthy	130 (60.75)	102.91 ± 7.99	1.30 (0.276)
	Not heathy	84 (39.25)	101.44 ± 8.88	

** *p* < 0.01. a: 20–29 age participants, b: 30–35 age participants, c: 36–40 age participants, d: >40 age participants.

**Table 2 ijerph-18-04781-t002:** Health promotion behaviors and related factors among dual working parents (*n* = 214).

Factors (Item Number)	Dual Working Parents
	M ± SD
Family cohesion (10)	40.6 ± 5.55
Positive psychological capital (18)	66.74 ± 10.78
Health promotion behaviors (29)	102.5 ± 8.56

**Table 3 ijerph-18-04781-t003:** Correlations between variables of a dual working parents (*n* = 214).

Category	Factors	Family Cohesion	Positive Psychological Capital	Health Promotion Behaviors
		r (*p*)	r (*p*)	r (*p*)
Dual working parents	Family cohesion	1		
	Positive psychological capital	0.37 (<0.0001)	1	
	Health promotion behaviors	0.31 (<0.0001)	0.32 (<0.0001)	1

**Table 4 ijerph-18-04781-t004:** Mediating effect of family cohesion in the relationship between positive psychological capital and health promotion behaviors (*n* = 214).

	Variables	Step 1	Step 2	Step 3
		Health Promotion Behaviors	Family Cohesion	Health Promotion Behaviors
		β (*p*)	β (*p*)	β (*p*)
Dual working parents	Intercept	85.38 (<0.001)	28.04881 (<0.001)	75.91 (<0.001)
	Positive psychological capital	0.26 (<0.001)	0.19 (<0.001)	0.19 (<0.001)
	Family cohesion			0.34 (0.002)
	R^2^	0.10	0.13	0.15
	F (*p*)	24.72 (<0.001)	32.62 (<0.001)	18.01 (<0.001)
	Sobel test: Z = 2.70, *p* = 0.005			

**Table 5 ijerph-18-04781-t005:** Moderating effect of family cohesion in the relationship between positive psychological capital and health promotion behaviors (n = 214).

	Variables	β	SE	t	*p*
Dual working parents	Intercept	72.64	8.32	8.73	<0.001
	Family cohesion	19.29	9.27	2.08	0.039
	Positive psychological capital	0.43	0.14	3.14	0.002
	Family × Positive psychological capital	−0.26	0.15	−1.73	0.084
		R^2^: 0.14 adjusted R^2^: 0.13 F: 11.59 *p*: < 0.001			

## Data Availability

The data that support the findings of this study are available from the corresponding author upon reasonable request.

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
