# Peer review of "Mediating and Moderating Effects of Family Cohesion between Positive Psychological Capital and Health Behavior among Early Childhood Parents in Dual Working Families: A Focus on the COVID-19 Pandemic"

_ijerph, 2021, doi:10.3390/ijerph18094781_

Round 1

Reviewer 1 Report

Great revisions to incorporate the spirit and intent of my previous recommendations to take your paper to the next level. By incorporating conflicting and divergent perspectives from the literature, updating your paper to note limitations to your paper; and undertaking an English, you have enhanced the comprehensiveness and readability of your research. I look forward to reading this current article, as well as other research you and your colleagues undertake in this area.

Reviewer 2 Report

The authors have made extensive revisions that improved the manuscript greatly. I congratulate them for diligent and careful work. I have no further suggestions or recommendations, beyond a careful reading for minor typos and very occasional awkward English language.

This manuscript is a resubmission of an earlier submission. The following is a list of the peer review reports and author responses from that submission.

Round 1

Reviewer 1 Report

IJERPH – Mediating and Moderating Effects of Family Cohesion Between Positive Psychological Capital and Health Behavior Among Early Childhood Parents in Dual Working Families: A 4 Focus on the COVID-19 Pandemic

IJERPH-1150844

Overall an interesting paper, which examined the mediating and moderating effects of family cohesion on the relationship between positive psychological capital and health promotion behaviors among dual working parents of young children during the COVID-19 pandemic. Given this research is focused on behaviors during the COVID 19 pandemic, the paper will garner interest from academia. From an academic perspective the mix of current (less than 5 year peer reviewed research) presented and discussed in the paper is generally robust in terms of currency, however, the paper does lack a discussion of divergent and conflicting perspectives, which is correctable with minor effort – its inclusion is necessary and appropriate since scholars expect journal articles which present recent research which also incorporates divergent and conflicting perspectives as opposed to a narrow focus which many academic would consider as inappropriate. On a positive note, the author(s) has provided some germane discussion with the literature – and briefing noting the divergent and conflicting findings and/or theoretical positions causing intellectual tension in the field would ensure the paper’s overall comprehensiveness and currency to the intended audience.

From a methodology perspective, the paper is well designed and appropriate – the exception is the lack of limitations at the front end of the paper; and this may very well reflect my own personal preferences. However, in defense of this posture, the identification of limitations at the front (Abstract and within the Methodology Section) enables the reader audience to understand the limitations and any caveats up front, and prior to reading the paper through and then findings limitations at the back end of the paper. It is imperative that reader(s) have the opportunity to place the paper into a context in relation to what is stated / proposed by the author.

The results are presented in a clear and concise manner; there is evidence of analysis evident; the inclusion of Tables and Figures are an effective visual. As well, the author(s) have developed an acceptable linkage between the results and conclusions noted. The points noted in paper are tied together into a final coherent picture. It is evident that the author(s) have an excellent understanding of the subject area.

This is a solid paper in many respects, since it provides several opportunities for continued research in the subject area with the possibility of different streams within the research area, while providing further avenues of research potential. With respect to the practical application of the research, it presents an opportunity to enhance the depth, breadth and understanding the mediating and moderating effects of family cohesion among parents, children and health promotion behaviors during the COVID-19 pandemic and how health care providers and governments can tailor strategies to enhance health behaviors among its populations.

A further opportunity is to have a professional edit of the paper since there are instances of grammatical, syntax errors which impacts flow and readability. So, the author(s) should consider a professional English edit

Confidential Comments to the Editor-in-Chief

Overall, this is a very good paper – however, as noted above there are some issues with respect to the absence of divergent and conflicting findings in the paper. As well, there is a need for the author(s) to incorporate limitations to the paper as noted above at the beginning of the paper, as opposed to the back end of the paper. Finally, the author(s) need to have a professional edit completed on the paper to ensure a readability and consistency of flow throughout the paper

Minor Revisions Required

Comments to the Author

A very good paper - For consideration, I recommend the inclusion of conflicting and divergent perspectives in the literature, and a further updated to the Abstract / Methodology to note the limitations to your paper as opposed to the back end of the paper. Finally, I would recommend a full professional English edit to support readability of your paper.

Reviewer 2 Report

This manuscript examines moderation and mediation effects of family cohesion, psychological capital and health promotion behavior among dual-working parents in South Korea. The sample size is substantial and the paper is very detailed. I have several comments for the authors to consider.

  1. The introduction and discussion are long and rather repetitive. Streamlined writing to make the key points might be feasible.
  2. I am a bit confused about the links to COVID-19 in this paper. From what I can tell, there were no measures of COVID influence on the family. Is that correct? Were any measures taken to see if family members became ill, if there was a death among friends or family, etc? How are the study measures relevant to COVID beyond the time when they were completed?
  3. Related to #2, I am confused by the conceptualization of this manuscript. It seems odd to hypothesize both mediational and moderational relations. What is the conceptual, theory-based hypothesis that was laid out before the study? This is not described well in the paper and we should have an introduction that leads into clear conceptualized hypotheses. A figure showing the hypothesized relationships might be useful to consider.
  4. The introduction spends a lot of time on COVID, but I was unclear whether the focus is on the medical consequences of COVID or the psychosocial factors associated with it. After reading the full paper, I remain confused and, in fact, feel as if the introductory descriptions of COVID are not particularly relevant to the goals of the study.
  5. If COVID remains a key part of the introduction, clarity concerning whether the focus is on acute infection issues, acute psychosocial issues, or chronic long-term psychosocial issues would be helpful.
  6. I also am confused about the time period of focus in both the introduction and conclusion. The COVID pandemic is rapidly moving and influencing us in rapidly-changing ways. When were these data collected and what was the state of COVID in South Korea during that time period? How might that influence interpretation of results, and how might that be meaningful for future implications?
  7. Line 44 – I am not sure “boredom” belongs in this list – feels out of place.
  8. Line 92 – I would argue schools, grandparents and childcare facilities are equally responsible for efforts to prevent disease, not just parents.
  9. It would be nice to have more clarification about the sample’s recruitment. Was the sample representative of a group of parents in South Korea? What is a Social Network System? How representative is the sample?
  10. Related to #9, it sounds like 233 parents consented. How many were contacted but did not complete the survey?
  11. Were all parents who participated from dual-working families? There is a suggestion that might be the case, especially in the tables, but it is never stated clearly.
  12. How was the health promotion measure modified (line 206)?
  13. Why was regression used instead of SEM?
  14. How did Table 1 adjust for 2-child households? The numbers don’t seem to add up correctly.
  15. What is the range of children?
  16. How is planned pregnancy, breastfeeding, and child health condition relevant to the hypotheses or conceptualization of the study?
  17. Table 2 lists subfactors of health promotion. Details on these should appear in the methods section. Were analyses conducted to determine if there were different patterns for different aspects of health promotion? Is this relevant?
  18. Figure 1 really presents the results from Table 4 effectively. Perhaps Table 4 is not needed, or at the least the authors might refer to Figure 1 earlier in the paper.
  19. Why are both moderation and mediation hypothesized? This reverts back to some of my earlier comments. Perhaps only mediation should be hypothesized, and the discussion of moderation dropped out of the paper?
  20. In the discussion section, the links to COVID-19 seem tenuous to me. I fail to see the connection between this study and COVID, except perhaps the timing of the survey. From what I can tell, no questions specific to COVID or the pandemic were asked of participants. Or did I misunderstand something?
  21. Lines 336-337, this sentence seems overstated and goes beyond the results of the study.
  22. Lines 427-428 – this sentence seems strange since no moderating effects were reported.